# Non-ST elevation acute coronary syndrome in patients aged 80 years or older in Vietnam: An observational study

Tan Van Nguyen[1,2], Khai Xuan Bui[1], Khuong Dang Tran[1], Duong Le[2], Tu Ngoc Nguyen[3]*

**1** Department of Geriatrics & Gerontology, University of Medicine and Pharmacy in Ho Chi Minh City, Ho Chi Minh City, Vietnam, **2** Department of Interventional Cardiology, Thong Nhat Hospital, Ho Chi Minh City, Vietnam, **3** Westmead Applied Research Centre, Faculty of Medicine and Health, The University of Sydney, Sydney, New South Wales, Australia

* ngoc.tu.nguyen@sydney.edu.au

## Abstract

### Background

There is limited evidence of non-ST elevation acute coronary syndrome (NSTE-ACS) in patients aged 80 or older in Vietnam.

### Aim

To describe the clinical characteristics of patients aged≥80 with NSTE-ACS in Vietnam, and to examine the effect of percutaneous coronary intervention (PCI) on adverse outcomes.

### Methods

Consecutive patients aged ≥80 with a diagnosis of NSTE-ACS admitted to two tertiary hospitals in Vietnam from 12/2018 to 06/2019 were recruited. The major outcomes were: (1) the composite of all-cause mortality, recurrent myocardial infarction and stroke, (2) re-admission rate during 3 months. Cox proportional-hazards regressions were conducted to examine the impact of PCI on the study outcomes, with results presented as hazard ratios (HR) and 95% confidence intervals (CI).

### Results

There were 120 participants, mean age 84.8 ± 3.8, 50% were female. Angiography and PCI were performed in 42 participants (35.0%). Most of the participants had multimorbidity and multiple coronary vessel disease. Compared to participants who did not receive PCI, participants who received PCI had significantly lower rates of adverse events during hospitalisation and during 3 months of follow up. Cox proportional hazards models adjusted to age and GRACE score show that PCI was significantly associated with reduced the composite outcome of all-cause mortality, recurrent myocardial infarction and stroke during 3 months follow-up (adjusted HR 0.32, 95%CI 0.12–0.86). PCI was also associated with reduced re-admission.

**Data Availability Statement:** All relevant data are within the manuscript and its Supporting Information files.

**Funding:** The author(s) received no specific funding for this work.

**Competing interests:** The authors have declared that no competing interests exist.

## Conclusions

The rate of PCI was low in the very elderly patients with NSTE-ACS in this study, although PCI was significantly associated with reduced adverse outcomes.

## Introduction

Ischemic heart disease is one of the leading cause of death worldwide, particularly in older people. [1] Patients with ischemic heart disease may present with chronic stable angina or acute coronary syndromes. [2] Acute coronary syndromes (ACS) include ST-segment elevation myocardial infarction (STEMI) and non-ST elevation ACS, which consists of non-ST elevation myocardial infarction (NSTEMI) and unstable angina. [2] The incidence of non-ST elevation ACS is increasing due to the successful application of preventive therapies such as aspirin, statins, smoking cessation, and due to the ageing population. [3] In the Global Registry of Acute Coronary Events conducted in 24165 with ACS patients from 102 hospitals in 14 countries, the incidence of NSTEMI increased with advanced age (41% in patients aged ≥85 years compared to 30% in those under 65 years old). [4]

In patients with non-ST elevation ACS, early invasive treatment including percutaneous coronary intervention (PCI) in the absence of contraindication is recommended for patients with high risk. [2] According to the 2018 ESC/EACTS Guidelines on myocardial revascularization, the decision-making process of PCI for non-ST elevation ACS depends on many factors, including clinical presentation, comorbidities, risk stratification, and other features such as estimated life expectancy, the functional and anatomical severity of the coronary arteries. [5] However, there is limited evidence on the treatment of non-ST elevation ACS in the very elderly patients. Patients aged 75 or older just accounted for under 10% of all patients enrolled in trials, and those aged 80 or older were usually excluded from clinical trials. [6] Reports of poor outcomes in the elderly from some studies have raised concern about the risk versus benefit of PCI in the treatment of ACS in this population. [3]

The world's population is rapidly aging. By 2050 one in five people will be over 60 years old, and the number of people aged over 80 is projected to triple from 143 million in 2019 to 426 million in 2050. [7] Over the past decades, the global burden of cardiovascular disease has shifted towards low- and middle-income countries. [1] Vietnam is a lower middle-income country in Southeast Asia with rapid urbanization and aging population. In Vietnam, the percentage of older people is significantly increasing, with an estimate of 26.1% people aged 60 plus and 4.2% people aged 80 or older in 2049. [8] Previous studies showed that cardiovascular disease is the leading cause of death in Vietnam. [9–12] The prevalence of risk factors for ischemic heart disease such as obesity, diabetes, low physical activity, high consumption of alcohol are increasing in Vietnamese people. [13] However, there is limited evidence on the management of ACS in the very older patients in this population.

Therefore, this study aims to describe the clinical characteristics of the very elderly patients with non-ST elevation ACS in Vietnam, and to examine the effect of percutaneous coronary intervention on adverse outcomes during 3 months follow up.

## Methods

### Participants

A prospective, observational cohort study was conducted in patients admitted with ACS to Thong Nhat Hospital in Ho Chi Minh City (Interventional Cardiology Department) and Cho

Ray Hospital (Cardiology Department) from 12/2018 to 06/2019. These are the two large tertiary hospitals in Ho Chi Minh City, Vietnam. Consecutive patients were recruited during the study period.

Patients aged ≥80 and were diagnosed with non-ST elevation ACS on admission were invited to participate in this study. We excluded patients who were having: (1) severe illness (dying or receiving intensive care), (2) blind or deaf, (3) severe dementia or delirium, and (4) were unable to speak or understand Vietnamese language.

The study was approved by the ethics committees of the University of Medicine and Pharmacy in Ho Chi Minh City (Reference Number 454/DHYD-HDDD). Written informed consent was obtained from all participants.

## Data collection

Data were collected from patient interviews and from medical records using a predefined data collection form. Information obtained from medical records were: demographic characteristics, height, weight, medical history, comorbidities, admission diagnosis, GRACE score [14], coronary angiography and PCI during hospitalization, length of stays, cardiovascular medications prescribed at admission and at discharge, and adverse events during hospitalization (all-cause mortality, recurrent myocardial infarction, stroke, major bleeding, heart failure).

All participants were followed up for 3 months after discharged. Phone calls were conducted to the phone numbers provided by participants or their family to obtain information about adverse events during the 3 months, including readmission, bleeding events, and mortality.

The study major outcomes were: (1) the composite of all-cause mortality, recurrent myocardial infarction and stroke during 3 months since recruited, and (2) re-admission due to cardiovascular causes during 3 months since recruited.

Sample size considerations: We estimated our study sample size based on the result of a study conducted in a cohort of elderly patients with ACS in Sweden. [15] In that study in 491 patients (mean age 83), the mortality rate after 1 year was 13% in the PCI group and 39.3% in the non-PCI group. Power analysis indicated that at least 43 participants would be needed in each group to detect the difference in mortality rate (at the power of 80% and p = 0.05).

## Statistical analysis

Analysis of the study data was performed with SPSS for Windows 24.0. Continuous variables are presented as mean ± standard deviation, and categorical variables as frequency and percentage. Comparisons between the two groups (PCI and non-PCI) were assessed using Chi-square tests for categorical variables and Student's t-tests or Mann-Whitney tests for continuous variables. Two-tailed P values < 0.05 were considered statistically significant.

To compare the time to the study major outcomes in participants with and without PCI, the Kaplan–Meier estimator was applied to compute survival curves over the 3-month follow-up period and differences between the two groups assessed using log rank tests.

Cox proportional-hazards regressions were conducted to examine the impact of PCI on the study major outcomes, with results presented as hazard ratios (HR) and 95% confidence intervals (CIs). All variables were examined for interaction and multicollinearity.

## Results

A total of 120 participants were recruited, mean age 84.8 ± 3.8, 50% were female, 72.5% had two or more chronic diseases. Of these, 42 participants (35.0%) underwent angiography and percutaneous coronary intervention. Overall, participants who received PCI had higher

**Table 1. Baseline characteristics of participants.**

| | All (N = 120) | Participants who did not receive PCI (N = 78) | Participants who received PCI (N = 42) | P |
|---|---|---|---|---|
| Age | 84.8 ± 3.8 | 85.2 ± 4.3 | 84.1 ± 2.6 | 0.152 |
| Female | 60 (50.0) | 43 (55.1) | 17 (40.5) | 0.126 |
| Smoking | 45 (37.5) | 27 (34.6) | 18 (42.9) | 0.374 |
| Multimorbidity (≥2 chronic diseases) | 87 (72.5) | 57 (73.1) | 30 (71.4) | 0.847 |
| Hypertension | 108 (90.0) | 67 (85.9) | 41 (97.6) | 0.041 |
| Diabetes | 27 (22.5) | 18 (23.1) | 9 (21.4) | 0.837 |
| Heart failure | 47 (39.2) | 38 (48.7) | 9 (21.4) | 0.003 |
| Stroke | 4 (3.3) | 3 (3.9) | 1 (2.4) | 0.670 |
| Chronic kidney disease | 24 (20.0) | 19 (24.4) | 5 (11.9) | 0.104 |
| Previous MI | 20 (16.7) | 12 (15.4) | 8 (19.1) | 0.608 |
| Previous PCI | 16 (13.3) | 9 (11.5) | 7 (16.7) | 0.431 |
| Previous CABG | 1 (0.8) | 0 (0) | 1 (2.4) | 0.171 |
| GRACE score | 162.8 ± 18.6 | 165.6 ± 20.7 | 157.6 ± 12.2 | 0.023 |
| Blood tests: | | | | |
| Creatinin (mg/dl) | 1.2 ± 0.4 | 1.2 ± 0.5 | 1.1 ± 0.3 | 0.040 |
| GFR (ml/m/1.73m$^2$) | 59.6 ± 18.8 | 56.5 ± 19.6 | 65.3 ± 16.1 | 0.014 |

Continuous data are presented as mean ± standard deviation; categorical data are shown as n (%). MI, myocardial infarction. PCI, percutaneous coronary intervention. CABG, coronary artery bypass grafting. GFR, glomerular filtration rate.

prevalence of hypertension, lower prevalence of heart failure history, lower GRACE score and lower serum creatinine level. There was no significant difference in age, sex, and the presence of multimorbidity between the two groups (Table 1).

The characteristics of coronary lesions and revascularization were described in Table 2. Most of the patients had multiple vessel disease (45.2% had 2 vessel disease, 33.3% had 3 vessel disease). Left anterior descending artery (LAD) stenosis was the most common (95.2%),

**Table 2. Coronary lesions and myocardial revascularisation in participants received angiography and PCI (N = 42).**

| | N = 42 |
|---|---|
| Total number of vessel disease | 2.1 ± 0.7 |
| No vessel disease | 0 (0) |
| 1 vessel disease | 9 (21.4) |
| 2 vessel disease | 19 (45.2) |
| 3 vessel disease | 14 (33.3) |
| Left main stenosis | 12 (28.6) |
| LAD stenosis | 40 (95.2) |
| RCA stenosis | 29 (69.0) |
| LCx stenosis | 20 (47.6) |
| PCI | 42 (100.0) |
| PCI within 24 hours since admission | 10 (23.8) |
| PCI at 24–72 hours since admission | 18 (42.9) |
| PCI after 72 hours since admission | 14 (33.3) |

Continuous data are presented as mean ± standard deviation; categorical data are shown as n (%). PCI, percutaneous coronary intervention. LAD, left anterior descending artery. RCA, right coronary artery. LCx, left circumflex artery.

**Table 3. Cardiovascular medication use at admission and at discharge.**

| | All (N = 120) | Participants who did not receive PCI (N = 78) | Participants who received PCI (N = 42) | P |
|---|---|---|---|---|
| **At admission:** | | | | |
| Enoxaparin | 90 (75.0) | 67 (85.9) | 23 (54.76) | <0.001 |
| Aspirin | 120 (100.0) | 78 (100.0) | 42 (100.0) | N/A |
| Clopidogrel | 100 (80.3) | 70 (89.74) | 30 (71.43) | 0.01 |
| Ticagrelor | 12 (10.0) | 0 (0.0) | 12 (28.6) | <0.001 |
| Statin | 113 (94.2) | 72 (98.3) | 41 (97.6) | 0.236 |
| Beta-blockers | 65 (54.2) | 39 (50.0) | 26 (61.9) | 0.212 |
| ACE inhibitors/ARBs | 98 (81.7) | 60 (76.9) | 38 (90.5) | 0.067 |
| Aldosterone antagonists | 63 (52.5) | 40 (51.28) | 23 (54.76) | 0.716 |
| Nitrate | 61 (50.8) | 39 (50.0) | 22 (52.4) | 0.803 |
| **At discharge:** | | | | |
| Aspirin | 114 (95.0) | 72 (92.3) | 42 (100.0) | 0.090 |
| Clopidogrel | 93 (77.5) | 63 (80.8) | 30 (71.4) | 0.243 |
| Ticagrelor | 12 (10.0) | 0 (0.0) | 12 (28.6) | <0.001 |
| Statin | 107 (89.2) | 66 (84.6) | 41 (97.6) | 0.032 |
| Beta-blockers | 83 (69.2) | 50 (64.1) | 33 (78.6) | 0.102 |
| ACE inhibitors/ARBs | 97 (80.8) | 59 (75.6) | 38 (90.5) | 0.055 |
| Aldosterone antagonists | 61 (50.8) | 38 (48.7) | 23 (54.8) | 0.528 |
| Nitrate | 58 (48.3) | 36 (46.2) | 22 (52.4) | 0.515 |

Continuous data are presented as mean ± standard deviation; categorical data are shown as n (%). PCI, percutaneous coronary intervention. ACE, Angiotensin-converting-enzyme. ARBs, angiotensin II receptor blockers.

followed by right coronary artery (RCA) (69.0%), and left circumflex artery (LCx) (47.6%). Left main stenosis was present in 28.6% of the participants.

Table 3 presents the use of cardiovascular medication during hospitalization and at discharge. Overall, upon discharge, the prescription of secondary prevention medications was high in patients aged 80 years or older with ACS (aspirin 95.0%, clopidogrel 77.5%, statins 89.2%, angiotensin converting enzyme (ACE) inhibitors/ angiotensin II receptor blockers (ARBs) 80.8%, beta-blockers 69.2%). The prescription of secondary prevention medications was similar in patients with and without PCI, except for statin (97.6% in patients with PCI compared to 84.6% in patients without PCI, p = 0.03).

Compared to participants who did not receive PCI, participants who received PCI had significantly lower rates of adverse outcomes during hospitalization and during 3 months of follow up. The rate of major bleeding was low during hospitalization (0.83% overall, 0% in the PCI group versus 1.3% in the non-PCI group, p = 0.461) and during 3 months follow up (2.5% overall, 4.8% in the PCI group versus 1.3% in the non-PCI group, p = 0.280). (Table 4)

The Kaplan-Meier survival function for composite outcome and readmission indicated that at all points in time during the three-month follow-up, participants who received PCI treatment were less likely to have the composite outcome (Log Rank Chi-Square 4.564, 1df, p = 0.033 and Breslow Chi-Square 5.000, 1df, p = 0.025) and less likely to be readmitted to hospitals (Log Rank Chi-Square 5.733, 1df, p = 0.017 and Breslow Chi-Square 6.138, 1df, p = 0.013) compared to those who did not receive PCI (Figs 1 and 2).

On univariate survival analyses, only PCI (unadjusted HR 0.27, 95%CI 0.10–0.70), age (unadjusted HR 1.15, 95%CI 1.05–1.26) and GRACE score (unadjusted HR 1.02, 95%CI 1.00–1.04) were significantly associated with the composite outcome. (Table 5) The relationship

**Table 4. Adverse outcomes of the participants during hospitalization and after 3 months of follow up.**

| | All (N = 120) | Participants who did not receive PCI (N = 78) | Participants who received PCI (N = 42) | P |
|---|---|---|---|---|
| Hospitalisation length (days) | 7.1 ± .1.5 | 7.7 ± 1.3 | 6 ± 1.2 | <0.001 |
| Adverse outcomes during hospitalisation: | | | | |
| All-cause mortality | 6 (5.0) | 6 (7.7) | 0 (0) | 0.064 |
| Recurrent myocardial infarction | 40 (33.3) | 34 (43.6) | 6 (14.3) | 0.001 |
| Stroke | 0 (0.0) | 0 (0) | 0 (0) | N/A |
| Congestive heart failure | 57 (47.5) | 46 (58.9) | 11 (26.2) | 0.001 |
| Major bleeding | 1 (0.83) | 1 (1.3) | 0 (0) | 0.461 |
| Adverse outcomes after 3 months: | | | | |
| Re-admission | 49 (43.0) | 37 (51.4) | 12 (28.6) | 0.018 |
| Composite outcomes | 34 (28.3) | 29 (37.2) | 5 (11.9) | 0.003 |
| All-cause mortality | 23 (19.2) | 20 (25.6) | 3 (7.1) | 0.015 |
| Recurrent myocardial infarction | 9 (7.5) | 8 (10.3) | 1 (2.4) | 0.158 |
| Stroke | 2 (1.7) | 1 (1.3) | 1 (2.4) | 1.000 |
| Major bleeding | 3 (2.5) | 1 (1.3) | 2 (4.8) | 0.280 |

Composite outcomes were defined as the combination of all-cause mortality, recurrent myocardial infarction, stroke. PCI, percutaneous coronary intervention.

between PCI and the composite outcome was still significant after adjusted for age and GRACE score in multivariate survival analysis (adjusted HR 0.32, 95%CI 0.12–0.86).

Cox proportional hazards models on PCI and other related factors on time to re-admission show that only PCI was significantly associated with 3-month re-admission (unadjusted HR 0.26, 95%CI 0.09–0.75). Multivariate survival analysis was not applied for 3-month re-admission as there were no other factors that show significant association with this outcome on univariate analysis (Table 5).

## Discussion

In this study in patients aged 80 years or older with non-ST elevation ACS, only around one third of the participants received PCI treatment. Overall, the prevalence of multimorbidity and the multiple coronary vessel disease was high in the participants. The prescription of secondary prevention medications at discharge was high in both groups of patients with and without PCI. PCI was significantly associated with reduced adverse outcomes during the 3 months follow up. The rate of major bleeding was low and there was no significant difference between the PCI group and the non-PCI group.

Our findings are compatible with other studies in this topic in the world. Older people with acute coronary syndrome usually have multiple chronic health conditions. [16] Previous studies showed that older patients are less likely to receive PCI, and coronary lesions in older people are usually complicated and involved multiple vessels. [3,4,16] In the Global Registry of Acute Coronary Events Study in 35512 patients with non-ST elevation ACS from 113 hospitals from 14 countries in North and South America, Europe, Australia and New Zealand, angiography was performed in 33% of the very elderly (compared to 67% in younger patients). [17]

In the past, the rate of complications after PCI in older people was higher compared to younger people, however, in recent years, due to advance in interventional cardiology, the number of older patients with non-ST elevation ACS that received PCI treatment is increasing. [3,18] Many studies have shown that PCI in older patients is effective in reducing adverse outcomes without increasing bleeding risk. According to the After Eighty Study, a randomized controlled trial in patients aged 80 years or older with non-ST elevation ACS (229 participants

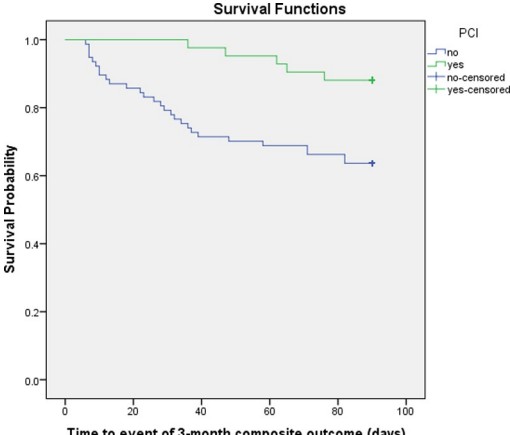

**Fig 1. The Kaplan-Meier survival curves for composite outcome after 3 months in participants with and without PCI.** PCI, percutaneous coronary intervention.

in the invasive group and 228 participants in the conservative group), an invasive strategy is superior to a conservative strategy in the reduction of composite events, and there was no difference in the rates of bleeding complications between the two strategies. [19] In the Acute Coronary Syndromes Registry study in 1936 patients ≥75 years with NSTEMI (1005 patients underwent coronary angiography and revascularization if indicated, and 931 patients received conservative treatment), in-hospital mortality and the combined outcome of mortality/non-fatal recurrent myocardial infarction were lower in patients receiving invasive management compared with those managed by conservative strategy (6.0% versus 12.5%, p<0.001 and 9.6% vs 17.3%, p<0.001, respectively). [20] There was also a significant reduction in 1-year mortality in the invasive treatment group compared to the conservative treatment group (OR 0.56, 95% CI 0.38–0.81). [20] In the Italian Elderly ACS study in 313 non-ST elevation ACS patients ≥75 years old, there was a significant reduction in the primary outcome (death, myocardial infarction, stroke, and readmission due to cardiovascular causes) in the early aggressive treatment group with elevated troponin on admission compared to the conservative group at 1 year (HR 0.43, 95% CI 0.23–0.80). [21] In the Alberta Provincial Project for Outcomes Assessment in

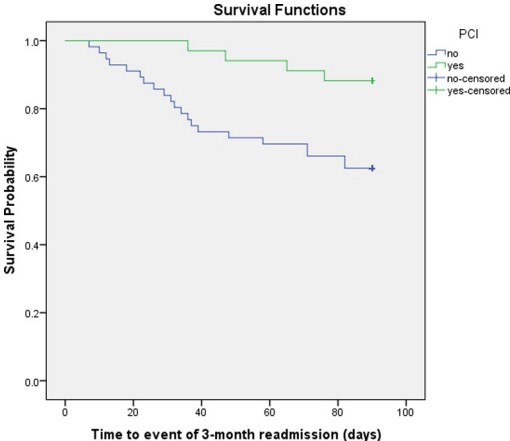

**Fig 2. The Kaplan-Meier survival curves for readmission after 3 months in participants with and without PCI.** PCI, percutaneous coronary intervention.

**Table 5. Cox proportional hazards model of PCI and other related factors on time to adverse outcomes after 3 months follow up in patients aged ≥80 with ACS.**

| Outcomes | Unadjusted HR (95%CI) for the composite outcome (N = 120) | P | Unadjusted HR (95%CI) for re-admission (N = 114) | P |
|---|---|---|---|---|
| PCI | 0.27 (0.10–0.70) | 0.007 | 0.26 (0.09–0.75) | 0.013 |
| Age | 1.15 (1.05–1.26) | 0.002 | 1.07 (0.96–1.19) | 0.250 |
| Female | 0.92 (0.46–1.82) | 0.807 | 0.90 (0.41–1.98) | 0.797 |
| GRACE score | 1.02 (1.00–1.04) | 0.025 | 1.00 (0.98–1.03) | 0.730 |
| Heart failure | 1.63 (0.82–3.23) | 0.160 | 1.64 (0.75–3.59) | 0.218 |
| Diabetes | 0.46 (0.16–1.31) | 0.147 | 0.34 (0.08–1.44) | 0.143 |
| Hypertension | 0.54 (0.21–1.39) | 0.198 | 0.66 (0.20–2.19) | 0.494 |
| Stroke | 2.05 (0.49–8.57) | 0.327 | 3.38 (0.79–14.48) | 0.101 |
| History of IHD | 1.30 (0.59–2.88) | 0.519 | 2.21 (0.95–5.13) | 0.064 |
| Chronic kidney disease | 1.33 (0.60–2.95) | 0.480 | 1.22 (0.46–3.26) | 0.687 |
| Multimorbidity (≥2 chronic diseases) | 0.64 (0.32–1.31) | 0.222 | 0.73 (0.32–1.66) | 0.455 |

PCI, percutaneous coronary intervention. IHD, ischemic heart disease.

Coronary Heart Disease (APPROACH), clinical data and outcomes of all patients undergoing cardiac catheterization and revascularisation in the province of Alberta, Canada since 1995 were recorded. They found that the absolute risk differences in comparison to medical therapy for PCI and coronary artery bypass grafting (CABG) were greater for patients aged 80 or older compared to younger patients. In 983 patients aged 80 plus, survival was 71.6% for PCI, 77.4% for CABG and 60.3% for medical therapy. [22]

In this study in the over 80s with ACS, we also found that the prevalence of prescription of secondary prevention medications was high and similiar between the PCI and non-PCI group (except for statin). This finding is compatible to a previous study conducted in 2013 at other hospitals in Vietnam, in which a high physicians' adherence to prescribing guidelines for ACS was reported. [23]

## Strength and limitations

To our best knowledge, this is the first study describing the clinical characteristics of patients aged 80 years or older with non-ST elevation ACS in Vietnam and investigating the effect of PCI on adverse outcomes in this population. This study was conducted at two large tertiary hospitals in Ho Chi Minh City, Vietnam and contained high quality detailed clinical information. The major limitation of this study is that the follow up time was short. The second limitation is that information on socioeconomics of the participants was not collected. In Vietnam, socioeconomic status may have a significant impact on the rate of PCI treatment in patients with ACS. Thirdly, this study was conducted in elderly patients at only two hospitals, which may not be representative for all older patients with non-ST elevation ACS in Vietnam.

## Implications for research and clinical practice

The clinical evidence for treatment of ACS in the elderly is less robust than in younger patients and in fact, there is no specific guidelines for the management of ACS in general and non-ST elevation ACS in particular in patients aged 80 years or older. In elderly patients, non-ST elevation ACS is more common than STEMI. [24] In Vietnam, there has been limited evidence on the epidemiology and treatment of non-ST elevation ACS in patients aged 80 years or older. This study suggests further research on this topic in Vietnam. Most of older people in

Vietnam (around 73%) are living in rural areas, where most hospitals do not have interventional facilities. [25] Evidence from large international cohort studies showed that the availability of interventional facilities at the admitting hospital is a major predictor of cardiac catheterization and PCI procedure. [26,27] While the recognition and treatment for STEMI is more straightforward, non-ST ACS may be more likely to be under-evaluated and under-treated, especially in rural settings. The findings from this study also suggest the need to increase awareness about the benefits of PCI for patients aged 80 years or older with non- ST elevation ACS in Vietnam, especially in rural clinical settings, and to develop strategies to ensure that elderly patients with non-ST elevation ACS who are candidates for PCI should be promptly referred to hospitals with interventional facilities.

## Conclusion

In this study in patients aged 80 years or older with non-ST elevation ACS in Vietnam, one third of the participants received PCI treatment and we found that PCI was significantly associated with reduced adverse outcomes during the study follow up time. The findings of this study contribute to the evidence of the benefits of PCI treatment in older patients with non-ST elevation ACS. Advanced aged alone should not be a contra-indication for PCI and all patients with non-ST elevation ACS should be evaluated and managed according to the current guidelines.

## Supporting information

**S1 Data.**
(SAV)

## Author Contributions

**Conceptualization:** Tan Van Nguyen.

**Data curation:** Tan Van Nguyen, Khai Xuan Bui.

**Formal analysis:** Tan Van Nguyen, Khai Xuan Bui, Khuong Dang Tran, Duong Le, Tu Ngoc Nguyen.

**Investigation:** Tan Van Nguyen, Khai Xuan Bui.

**Methodology:** Tan Van Nguyen, Khai Xuan Bui, Khuong Dang Tran, Duong Le, Tu Ngoc Nguyen.

**Project administration:** Tan Van Nguyen, Khai Xuan Bui, Khuong Dang Tran, Duong Le.

**Resources:** Tan Van Nguyen.

**Software:** Tan Van Nguyen, Tu Ngoc Nguyen.

**Supervision:** Tan Van Nguyen.

**Validation:** Tan Van Nguyen, Tu Ngoc Nguyen.

**Visualization:** Tan Van Nguyen, Khai Xuan Bui, Khuong Dang Tran, Duong Le.

**Writing – original draft:** Tan Van Nguyen, Khai Xuan Bui, Tu Ngoc Nguyen.

**Writing – review & editing:** Tan Van Nguyen, Khai Xuan Bui, Khuong Dang Tran, Duong Le, Tu Ngoc Nguyen.

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
