## [Decision Letter · Decision Letter 0]

16 Apr 2020

PONE-D-19-32737

Non-ST elevation acute coronary syndrome in patients aged 80 years or older in Vietnam: an observational study

PLOS ONE

Dear Dr Nguyen,

Thank you for submitting your manuscript to PLOS ONE. After careful consideration, we feel that it has merit but does not fully meet PLOS ONE’s publication criteria as it currently stands. Therefore, we invite you to submit a revised version of the manuscript that addresses the points raised during the review process.

We would appreciate receiving your revised manuscript by May 31 2020 11:59PM. To enhance the reproducibility of your results, we recommend that if applicable you deposit your laboratory protocols in protocols.io, where a protocol can be assigned its own identifier (DOI) such that it can be cited independently in the future. For instructions see: http://journals.plos.org/plosone/s/submission-guidelines#loc-laboratory-protocols

We look forward to receiving your revised manuscript.

Kind regards,

Chiara Lazzeri

Academic Editor

PLOS ONE

Journal Requirements:

1. Please ensure that you refer to Figure 2 in your text as, if accepted, production will need this reference to link the reader to the figure.

Reviewers' comments:

Reviewer's Responses to Questions

**Comments to the Author**

1. Is the manuscript technically sound, and do the data support the conclusions?

Reviewer #1: Partly

Reviewer #2: Yes

2. Has the statistical analysis been performed appropriately and rigorously? 

Reviewer #1: No

Reviewer #2: Yes

3. Have the authors made all data underlying the findings in their manuscript fully available?

Reviewer #1: Yes

Reviewer #2: Yes

4. Is the manuscript presented in an intelligible fashion and written in standard English?

Reviewer #1: Yes

Reviewer #2: Yes

5. Review Comments to the Author

Reviewer #1: Interesting paper entitled "Non-ST elevation acute coronary syndrome in patients aged 80 years or older in

Vietnam: an observational study" where the authors state that sample size was not calculated as major limitation. This element limits unfortunately any interpretation of the data collected

Reviewer #2: This paper by Tan Van Nguyen et al. was aimed to assess the impact of PCI in ACS patients aged > 80ys.

This was a non randomized prospective study enrolling 120 patients, among those 42 participants (35.0%) were treated with invasive strategy. At three months follow up patients who received PCI had significantly lower rates of adverse outcomes.

This study has several limitations:

- The sample size is quite small to assess hard end points (mortality and cardiovascular events)

- The authors did not mention the reason why patients were treated with invasive or medical treatment

- Baseline condition of patient treated with PCI or medical therapy were different, with better status of patient treated with PCI

- It is already known that older patients with ACS treated with invasive strategy may benefit even more that young patients from guidelines indicated PCI. I believe the authors should focus more on the novelty of implementing a national system for invasive treatment of ACS also in mid- low income countries.

6. PLOS authors have the option to publish the peer review history of their article (what does this mean?). If published, this will include your full peer review and any attached files.

Reviewer #1: No

Reviewer #2: No

---

## [Author Response · Author response to Decision Letter 0]

18 Apr 2020

18th April 2020

To Dr Chiara Lazzeri

RE: RESPONSE TO REVIEWERS COMMENTS ON MANUSCRIPT ID PONE-D-19-32737

We would like to thank the reviewers for the time spent reviewing our manuscript and for the useful comments, and to thank the editors for the opportunity to respond.

We feel that the suggestions have strengthened the manuscript and have tried to address as many of the suggestions as possible as detailed below. We would be happy to address any further issues if required.

Reviewers' comments:

Reviewer's Responses to Questions

Comments to the Author

1. Is the manuscript technically sound, and do the data support the conclusions?

Reviewer #1: Partly

Reviewer #2: Yes

2. Has the statistical analysis been performed appropriately and rigorously?

Reviewer #1: No

Reviewer #2: Yes

3. Have the authors made all data underlying the findings in their manuscript fully available?

Reviewer #1: Yes

Reviewer #2: Yes

4. Is the manuscript presented in an intelligible fashion and written in standard English?

Reviewer #1: Yes

Reviewer #2: Yes

5. Review Comments to the Author

Reviewer #1: Interesting paper entitled "Non-ST elevation acute coronary syndrome in patients aged 80 years or older in Vietnam: an observational study" where the authors state that sample size was not calculated as major limitation. This element limits unfortunately any interpretation of the data collected

Response: We have added our justification for sample size. Please see lines 105-110:

“Sample size considerations: We estimated our study sample size based on the result of a study conducted in a cohort of elderly patients with ACS in Sweden.14 In that study in 491 patients (mean age 83), the mortality rate after 1 year was 13% in the PCI group and 39.3% in the non-PCI group. Sample size calculations indicated that at least 43 participants would be needed in each group to detect the difference in mortality rate (at the power of 80% and p=0.05).”

Reviewer #2: This paper by Tan Van Nguyen et al. was aimed to assess the impact of PCI in ACS patients aged > 80ys.

This was a non randomized prospective study enrolling 120 patients, among those 42 participants (35.0%) were treated with invasive strategy. At three months follow up patients who received PCI had significantly lower rates of adverse outcomes.

This study has several limitations:

- The sample size is quite small to assess hard end points (mortality and cardiovascular events)

Response: We have added our justification for sample size. Please see lines 105-110.

- The authors did not mention the reason why patients were treated with invasive or medical treatment.

Response: We have added the information for PCI decision in patients with non-ST elevation ACS as follows: (please see lines 59-63)

“According to the 2018 ESC/EACTS Guidelines on myocardial revascularization, the decision-making process of PCI for non-ST elevation ACS depends on many factors, including clinical presentation, comorbidities, risk stratification, and other features such as estimated life expectancy, the functional and anatomical severity of the coronary arteries”

- Baseline condition of patient treated with PCI or medical therapy were different, with better status of patient treated with PCI

Response: There was no significant difference in age, sex, and the presence of multimorbidity between patients who did and did not receive PCI. Patients with PCI had higher prevalence of hypertension, lower prevalence of heart failure, lower GRACE score and lower serum creatinine level. As patients treated with PCI had slightly better status, we did examine the influence of these factors in univariate and multivariate survival analysis. (Please see Table 5)

- It is already known that older patients with ACS treated with invasive strategy may benefit even more that young patients from guidelines indicated PCI. I believe the authors should focus more on the novelty of implementing a national system for invasive treatment of ACS also in mid- low income countries.

Response: Thank you for your suggestion. We have added a section of implications as follows (please see lines 223-239):

“Implications for research and clinical practice

The clinical evidence for treatment of ACS in the elderly is less robust than in younger patients and in fact, there is no specific guidelines for the management of ACS in general and non-ST elevation ACS in particular in patients aged 80 years or older. In elderly patients, non-ST elevation ACS is more common than STEMI.24 In Vietnam, there has been limited evidence on the epidemiology and treatment of non-ST elevation ACS in patients aged 80 years or older. This study suggests further research on this topic in Vietnam. Most of older people in Vietnam (around 73%) are living in rural areas, where most hospitals do not have interventional facilities.25 Evidence from large international cohort studies showed that the availability of interventional facilities at the admitting hospital is a major predictor of cardiac catheterization and PCI procedure. 26,27 While the recognition and treatment for STEMI is more straightforward, non-ST ACS may be more likely to be under-evaluated and under-treated, especially in rural settings. The findings from this study also suggest that there is a need to increase awareness about the benefits of PCI for patients aged 80 years or older with non-ST elevation ACS in Vietnam, especially in rural clinical settings, and to develop strategies to ensure that elderly patients with non-ST elevation ACS who are candidates for PCI should be promptly referred to hospitals with interventional facilities. ”

6. PLOS authors have the option to publish the peer review history of their article (what does this mean?). If published, this will include your full peer review and any attached files.

Do you want your identity to be public for this peer review? For information about this choice, including consent withdrawal, please see our Privacy Policy.

Reviewer #1: No

Reviewer #2: No

Journal Requirements:

1. Please ensure that you refer to Figure 2 in your text as, if accepted, production will need this reference to link the reader to the figure.

Response: We have added the reference linked to Figure 2 in the main text (Please see line 148)

2. Please include captions for your Supporting Information files at the end of your manuscript, and update any in-text citations to match accordingly. Please see our Supporting Information guidelines for more information: https://protect-au.mimecast.com/s/U67cCwV1vMfP99JvH1Sq19?domain=journals.plos.org.

Response: We have added the caption (Please see line 260)

---

## [Editor Report · Decision Letter 1]

4 May 2020

Non-ST elevation acute coronary syndrome in patients aged 80 years or older in Vietnam: an observational study

PONE-D-19-32737R1

Dear Dr. Nguyen,

We are pleased to inform you that your manuscript has been judged scientifically suitable for publication and will be formally accepted for publication once it complies with all outstanding technical requirements.

With kind regards,

Chiara Lazzeri

Academic Editor

PLOS ONE
---

## [Editor Report · Acceptance letter]

2 Jun 2020

PONE-D-19-32737R1 

Non-ST elevation acute coronary syndrome in patients aged 80 years or older in Vietnam: an observational study 

Dear Dr. Nguyen:

I'm pleased to inform you that your manuscript has been deemed suitable for publication in PLOS ONE. Congratulations! Your manuscript is now with our production department. 

Kind regards, 

on behalf of

Dr. Chiara Lazzeri 

Academic Editor

PLOS ONE